# Episome partitioning and symmetric cell divisions: Quantifying the role of random events in the persistence of HPV infections

Thomas Beneteau👁*, Christian Selinger👁©, Mircea T. Sofonea👁©, Samuel Alizon👁©

Laboratoire MIVEGEC, Université de Montpellier, Centre national de la recherche scientifique, Institut de recherche pour le développement, Montpellier, France

© These authors contributed equally to this work.

* thomas.beneteau@ird.fr

**Data Availability Statement:** All data and code used for running experiments, model fitting, and plotting is available on a gitlab repository at https://gitlab.in2p3.fr/ete/hpv-dynamic.

## Abstract

Human Papillomaviruses (HPV) are one of the most prevalent sexually transmitted infections (STI) and the most oncogenic viruses known to humans. The vast majority of HPV infections clear in less than 3 years, but the underlying mechanisms, especially the involvement of the immune response, are still poorly known. Building on earlier work stressing the importance of randomness in the type of cell divisions in the clearance of HPV infection, we develop a stochastic mathematical model of HPV dynamics that combines the previous aspect with an explicit description of the intracellular level. We show that the random partitioning of virus episomes upon stem cell division and the occurrence of symmetric divisions dramatically affect viral persistence. These results call for more detailed within-host studies to better understand the relative importance of stochasticity and immunity in HPV infection clearance.

## Author summary

Every year, infections by Human Papillomaviruses (HPV) are responsible for a large share of infectious cancers. The prevalence of HPVs is very high, which makes it a major public health issue. Fortunately, most HPV infections (80 to 90%) are cleared naturally within three years. Among the few that persist into chronic infections, the majority also naturally regress. Hence for a given HPV infection, the risk of progression towards cancerous status is low. The immune response is often invoked to explain HPV clearance in non-persisting infections, but many uncertainties remain. Besides immunity, randomness was also suggested to play an important role. Here, we examine how random events occurring during the life cycle of the virus could alter the persistence of the virus inside the host. We develop a mechanistic model that explicitly follows the dynamic of viral copies inside host cells, as well as the dynamics of the epithelium. In our model, infection extinction occurs when all viral copies end up in differentiated cells and migrate towards the surface. This can happen upon cell division during the random allocation of the episomes (i.e. independent circular DNA copies of the viral genome) or when a stem cell divides symmetrically to

**Funding:** SA acknowledges the support from European Research Council (ERC) under the European Union's Horizon 2020 research and innovation program (EVOLPROOF, grant agreement No 648963). TB is funded by a doctoral fellowship from the Ligue Contre le Cancer (grant No TAKX21133). The funders had no role in study design, data collection and analysis, decision to publish, or preparation of the manuscript.

**Competing interests:** The authors have declared that no competing interests exist.

generate two differentiated cells. We find that the combination of these random events drastically affects infection persistence. More generally, the importance of random fluctuations could match that of immunity and calls for further studies at the within-host and the epidemiological level.

## Introduction

Human Papillomaviruses (HPV) are the most oncogenic viruses known to humans [1]. Up to 8.6% of all cancers occurring worldwide in women are caused by HPV infections, versus 0.8% for cancers in men [2]. Among the various types of cancers attributable to HPV, cervical cancers stand out since they are the third-most diagnosed form of cancer in women [3].

The burden HPVs impose on human populations largely originates from their high prevalence in the general population. For example, it was estimated that the average lifetime probability (from sexual debut to age 70) of acquiring a genital HPV infection in the US was 84.6% for women and 91.3% for men [4]. But since up to 90% of these infections remain subclinical and are cleared in less than 3 years [5], viral persistence, in particular for specific genotypes such as HPV16, determines potential oncogenicity [6].

Despite its ubiquity, the mechanisms underlying HPV infection clearance and persistence are still poorly understood. Although the innate immune response (e.g. natural killer cells) is known to be involved in chronic infections [7], few data exist for non-persistent infections. Adaptive immune responses (e.g. the role of T helper cells) have been studied in the context of vaccine trials, but the results remain inconsistent [8]. It has been noticed that people with recurrent genital warts were more likely to be HPV seropositive than those who do not have a history of genital warts [9]. However, more generally, the breadth and magnitude of adaptive immune responses following HPV infection are believed to be limited [10] due to innate immune evasion mechanisms and a lack of inflammatory responses elicited by infected epithelial cells. Furthermore, in natural infections, there is no viremia (*i.e.* presence of viruses in the blood), and the production of detectable antibodies against the virus is low [10]. Another longitudinal study investigated HPV16 seroconversion statuses among various groups of infected women but did not reach any conclusive result, potentially by lack of statistical power [11].

The inherently stochastic nature of the episomal life cycle of HPV, i.e. replication of the viral genome as independent circular DNA copies, could be an additional factor contributing to the clearance of HPV infections. HPVs are non-lytic viruses that persist in the basal stem cells by replicating with host cell chromosomes [12]. In the basal layer, most cell divisions are asymmetric: one basal stem cell gives birth to one daughter stem cell, and one daughter differentiated cell upon cell division. Occasionally, especially in the context of a perturbation such as wounding [13, 14], symmetric cell divisions occur that either yield two stem cells or two differentiated cells. Infection persistence in the epithelial tissue occurs when viral copies are transmitted to the daughter stem cell during cell divisions. Conversely, transmission to new hosts occurs via the viral copies in the daughter differentiated cell. These copies migrate and maturate with the cell before being released to the surface upon shedding.

Symmetric divisions of infected basal stem cells into two daughter stem cells or two daughters differentiated cells [13, 15] strongly impact the spread and persistence of the virus in the basal layer of the tissue. Since HPV infections begin with few infected cells [16], randomness in the type of cell division plays a crucial role in the clearance of HPV infections [17]. A second source of stochasticity occurs within the nucleus of infected cells. During stem cell divisions, the number of episomes that are passed on to each daughter cell is random. Since cells are

infected with a small number (potentially unique) of viral copies [18], stochasticity in episome distribution upon division could impact the persistence of HPV infection. The partition of viral plasmids in human cells is uncommon but not specific to HPV. This phenomenon is still not clearly understood despite its importance in the maintenance of infections and progression towards cancers [19]. To date, this aspect of HPV infections has not been taken into account. In a recent review [20] on the use of mechanistic mathematical models applied to HPV infection, the authors globally call for more within-host modeling studies. While health-economic models have been extensively used to evaluate the cost-benefit of public interventions in the prevention of HPV infection or cervical cancer, mechanistic models have mainly been used to study some specific topics (e.g. viral evolution induced by vaccine use [21, 22] or impact of cell division in HPV natural infection [17]).

Building on the work of Ryser [17] that emphasized the contribution of symmetric cell divisions on HPV clearance, we develop an original model of HPV viral copies dynamic in epithelial basal layers.

Taken separately, the intra- and inter-cellular stochastic processes fail to capture part of the life cycle of HPV infections. The intracellular approach does not account for the spread of the infection inside the epithelial tissue, whereas the inter-cellular approach ignores the necessity of viral persistence in proliferating cells. Thus, we hypothesize that combining the two aspects can provide additional insights into the dynamics of HPV infection.

We investigate the role of stochasticity in the clearance of HPV infections and quantify the role of both symmetric divisions and episomal partitioning in this process. We provide further evidence for the crucial effect of randomness in the clearance of HPV infections.

## Methods

Our model focuses on the basal layer of a stratified squamous epithelium (SSE), but it can easily be adapted to other tissues as the HPV replication pattern is essentially the same in most sites [23, 24]. We only follow the dynamics of episomes in infected stem cells because SSE is in perpetual renewal and the turnover interval of the non-dividing cells lasts 3 to 6 weeks [25–27]. Therefore, in the absence of cell-to-cell transmission, it is improbable that an infection could persist only in the differentiated layers of SSE. We also neglect potential re-infection of the SSE from virions released by desquamating cells because these are thought to be unlikely given the importance of micro-wounding in the establishment of new infections [12].

The life-cycle is divided into two steps following the non-lytic properties of HPV: 1) during the division of an infected cell, HPV episomes are distributed randomly between the two daughter cells, and 2) infected stem cells can divide symmetrically, thereby ending or, conversely, further spreading the infection. We graphically summarize each aspect of the model in the S1 Fig. Both the intra- and inter-cellular mechanisms are modelled as discrete-time branching processes [28, 29].

More formally, let $X_n, n \in \mathbb{N}$ be the total number of HPV episomes in all infected stem cells, $n$ being the number of stem cell generations since the beginning of the infection. Unless stated otherwise, a time step corresponds to one cell generation. $X_n$ is a random variable, the distribution of which is non-trivial, for it is constructed following several random events. We assume the random events between cell lineages, *i.e.* descending cells by asymmetric divisions, to be independent and identically distributed. Within a cell lineage, we assume other stochastic events to be identically distributed (but not independent). Independence is verified under certain specific conditions, for instance if we assume a cell lineage only divides asymmetrically.

## Intracellular dynamics: Viral distribution during cell division

HPVs are non-lytic viruses that follow the life cycle of their host cells. In basal (stem) cells, the number of viral copies is usually considered to be limited to several dozens of episomes per cell: 10 to 200 according to Doorbar [16], and 50 to 100 according to more recent work [12]. We denote by $C$ the maximum number of episomes per infected stem cell. We model intracellular dynamics as the superposition of two distinct random steps:

1. A division, where each episome in the dividing cell is distributed to the two daughter cells according to a Bernoulli distribution *Bernoulli*($p$). For asymmetric divisions, parameter $p$ stands for the probability for an episome to be relocated in the daughter stem cell. Episomes distributed to the differentiated daughter cell are considered withdrawn from the system as they migrate to the surface without contributing to the infection of stem cells.

2. An amplification, which occurs once the division over. For this random phenomenon we either assume a Dirac distribution $\delta(\lambda)$ or a Poisson distribution *Poisson*($\lambda$), with $\lambda \in \mathbb{N}$. We refer to $\lambda$ as the episome replication factor.

Without prior knowledge of the biology of the amplification phase, the Dirac distribution appears to be the most parsimonious as it implicitly assumes that all viral copies exhibit the same average behavior. However, this assumption could be biased at the beginning of the infection as viruses remain in small numbers. In this case, the number of copies amplified could diverge from the average behavior due to individual variance. As little information is available in the literature, we choose a Poisson distribution as it only requires the same parameter as for the Dirac case.

The order of the two steps can be reversed without affecting the intrinsic characteristics of this branching process (see S1 Text for more details). Since HPV genomes are present in low copy numbers and HPV gene expression is set at a minimum level in basal stem cells [24], we assume that the number of HPV viral copies does not affect the dynamics of the host stem cells.

The infection can spread in the basal layer following symmetric divisions in two stem cells, a phenomenon governed by strong stochastic effects and detailed in the next section. To distinguish between daughter stem cells, we use the notation described in the supporting information S1 Text and define $p$ as the probability for an episome to be distributed in the left (**L**) daughter stem cell. We assume episome distribution bias towards one of the daughter cells does not apply when the two daughter cells are stem cells, hence we assume distribution to be even ($p = 0.5$) during symmetric division.

## Intra-host episome dynamics

In what follows, we group all the layers apical to the basal layer in a general population of differentiated cells. Let $(\Gamma_n)_{n \in \mathbb{N}}$ be the number of infected stem cells at time $n$. We denote by $s$ resp. $r$ the probability of a symmetric division to yield two stem cells resp. two differentiated cells. We assume $s$ and $r$ to be small compared to the probability of asymmetric cell division. Following the work of Ryser [17], we model infected stem cells dynamics using the following branching process:

$$\Gamma_{n+1} = \sum_{i=1}^{\Gamma_n} \phi_i \tag{1}$$

where $\phi_i$ are independent and identically distributed random variables with the following

distribution:

$$\phi_i = \begin{cases} 2 & \text{with probability } s \\ 0 & \text{with probability } r + \psi \\ 1 & \text{with probability } 1 - r - s - \psi \end{cases} \quad (2)$$

where $\psi$ is a random variable accounting for intracellular effects leading to the end of the infection of the infected stem cell $S$. A histological study from 1970 estimated that $r + s \leqslant 0.08$ [25], and a more recent analysis also estimated 8% of cell divisions to be symmetric [15].

## Numerical simulations and sensitivity analysis

Some analytical insights can be obtained from the previous equations but in-depth investigations require numerical simulations. We estimated the cumulative probability of extinction ($p_{ext}$) given a set of parameters (see Table 1 for further details on the parameters used) by carrying out $\Omega = 10^3$ independent runs parametrized by the same parameter values. We also include the type of intracellular amplification in the analysis (Dirac or Poisson distributed). We displayed a random trajectory of episomal dynamic in the S2 Fig.

Using the $\Omega$ different trajectories, we calculate the cumulative probability of extinction $p_{ext}(t)$ as follows:

$$p_{ext}(t, \mathcal{S}) = \frac{1}{\Omega} \sum_{k=1}^{\Omega} \mathbb{1}_{\{X_t^k = 0 | \mathcal{S}\}} \quad (3)$$

where $X_t^k | \mathcal{S}$ is the total number of episomes at time $t$ given parameters set $\mathcal{S}$ for the $k$-th trajectories. If $\Gamma_0 > 1$, we assume that all infected cells initially harbor $N_0$ viral copies, hence a total of $X_0 = \Gamma_0 \times N_0$ episomes. $p_{ext}$ is computed for all $t \leqslant 100$ cell generations. Based on data from the 1970s [25], we bound the rate of divisions between $[0.03, 0.07]$ day$^{-1}$, which means the upper bound of 100 cell divisions is sufficient to estimate infection clearance after 3 years. For each $t \in \mathbb{N}$, we estimate $p_{ext}$ using Maximum Likelihood estimation for a Binomial proportion. For each of these proportion we computed the 95% confidence interval using Agresti-Coull method [31].

To evaluate the role of each parameter on the probability of extinction, we perform a global sensitivity analysis using variance-based methods (also known as Sobol indices). This approach decomposes the variance of the simulation outputs (here $p_{ext}$) into fractions that can

**Table 1. Parameters notation and description along with their biologically estimated ranges.** The parameters $p$, $\lambda$, $C$ and $N_0$ govern the intracellular process, the three others are involved in the inter-cellular approach. Another parameter (not numerical) can be added to the list: the type of intracellular amplification (Dirac or Poisson distributed) as the two types slightly differ in their construction. Extensive analysis of the impact of each distribution on the probability of extinction is shown in S1 Text.

| Notation | Description | Range | Source |
|---|---|---|---|
| $p$ | Probability for an episome to be distributed to the daughter stem cell | [0, 1] | [30] |
| $\lambda$ | Episome replication factor (number of viral copies) | [2, 10] | — |
| $s$ | Probability of symmetric divisions into two stem cells | [0.01, 0.04] | [15, 25] |
| $r$ | Probability of symmetric divisions into two differentiated cells | [0.01, 0.04] | [15, 25] |
| $C$ | Viral capacity per infected stem cell | [10, 200] | [12, 16] |
| $N_0$ | Number of episome in each infected cells at the beginning of the epidemic | [1, C] | — |
| $\Gamma_0$ | Number of infected stem cells at the beginning of the epidemic | $\geqslant 1$ | — |

be attributed to each of our parameters or interactions between them [32, 33]. To do so, we store parameters sets in two different matrices $A$ and $B$ of dimension $n \times 7$, where the columns correspond to the six parameters $p$, $\lambda$, $s$, $r$, $C$, and $N_0$ (see Table 1) and the type of intracellular amplification. We denote by $A_B^{(k)}$ the matrix $A$ whose $k$-th column has been replaced by the $k$-th column of matrix $B$. We generate $A$ and $B$ sets using the Latin Hypercube Sampling (LHS) method implemented in the `lhs` package [34] in R v4.0.3 [35]. Each matrix contains $n = 500$ different parameter sets (more details can be found in S1 Text). We execute the global sensitivity analysis using `multisensi` package [36]. We compute model outputs (i.e. extinction probabilities) for all sets $A$, $B$, $A_B^{(k)}, k \in [\![1, 7]\!]$ to obtain the following Monte-Carlo estimator for the variance of the extinction probability up to time $t$ [37]:

$$\mathrm{Var}_i(t) = \frac{1}{\Omega} \sum_{k=1}^{\Omega} p_{ext}(t, B)_k \left( p_{ext}(t, A_B^{(k)}) - p_{ext}(t, (A)_k) \right)$$

We do not include the initial number of infected stem cells ($\Gamma_0$) in this analysis, since the sensitivity of $p_{\text{ext}}$ with respect to $\Gamma_0$ follows from the branching process property:

$$\mathbb{P}[X_t = 0 | \Gamma_0 = v] = \mathbb{P}[X_t = 0 | \Gamma_0 = 1]^v \tag{4}$$

## Results

Our model contains two nested branching processes, which makes it difficult to obtain analytical results. However, analytical expressions can be derived if we neglect either of the two stochastic processes (see S1 Text for detailed calculus and S3 and S4 Figs for numerical verification) and we use these results as boundaries to interpret the results of our numerical simulations and to determine how the cumulative probability of extinction ($p_{\text{ext}}$) varies for the different parameters. Simulations also allow us to explicitize this direction of variation in $p_{\text{ext}}$ with each parameter. Fig 1 displays the difference between probabilities of extinction for two parameter sets that only differ by the value of one parameter, we subtract the probability of extinction for the lowest of the two varying parameters to the probability of extinction of the highest of the two.

### Intracellular stochasticity

If stochasticity only acts at the within-cell level, *i.e.* symmetric divisions are assumed not to occur, we retrieve classical results from Bienaymé-Galton-Watson (BGW) branching process theory [29]: the extinction probability $p_{\text{ext}}$ then decreases with the episome replication factor, $\lambda$, and with the probability for an episome to end up in the daughter stem cell, $p$ (see Fig 2 for the Poisson case and S5 Fig for the Dirac distribution). As $\lambda$ increases, we observe that $p_{\text{ext}}$ converges toward $1 - p$. Biologically, this means that the main event governing the probability of extinction is the first cell division and the distribution of the original episome into a differentiated or a stem cell.

Numerical simulations allow us to investigate the effect of intracellular processes while accounting for symmetric divisions. In particular, our sensitivity analysis highlights the effect of $\lambda$ and $p$ (Fig 3). At the beginning of the infection, intracellular phenomena explain around 70% of the variance, mostly through the probability of allocation to daughter stem cell $p$. Indeed, in the early phase of the infection, the total number of episomes is usually low and, since symmetric divisions are rare ($r + s \leqslant 0.08$ [25]), the early dynamic of episomes is mostly governed by intracellular processes. We also retrieve the results from the analytical model, although the decrease in $p_{\text{ext}}$ with increasing $\lambda$ is more limited (Fig 1).

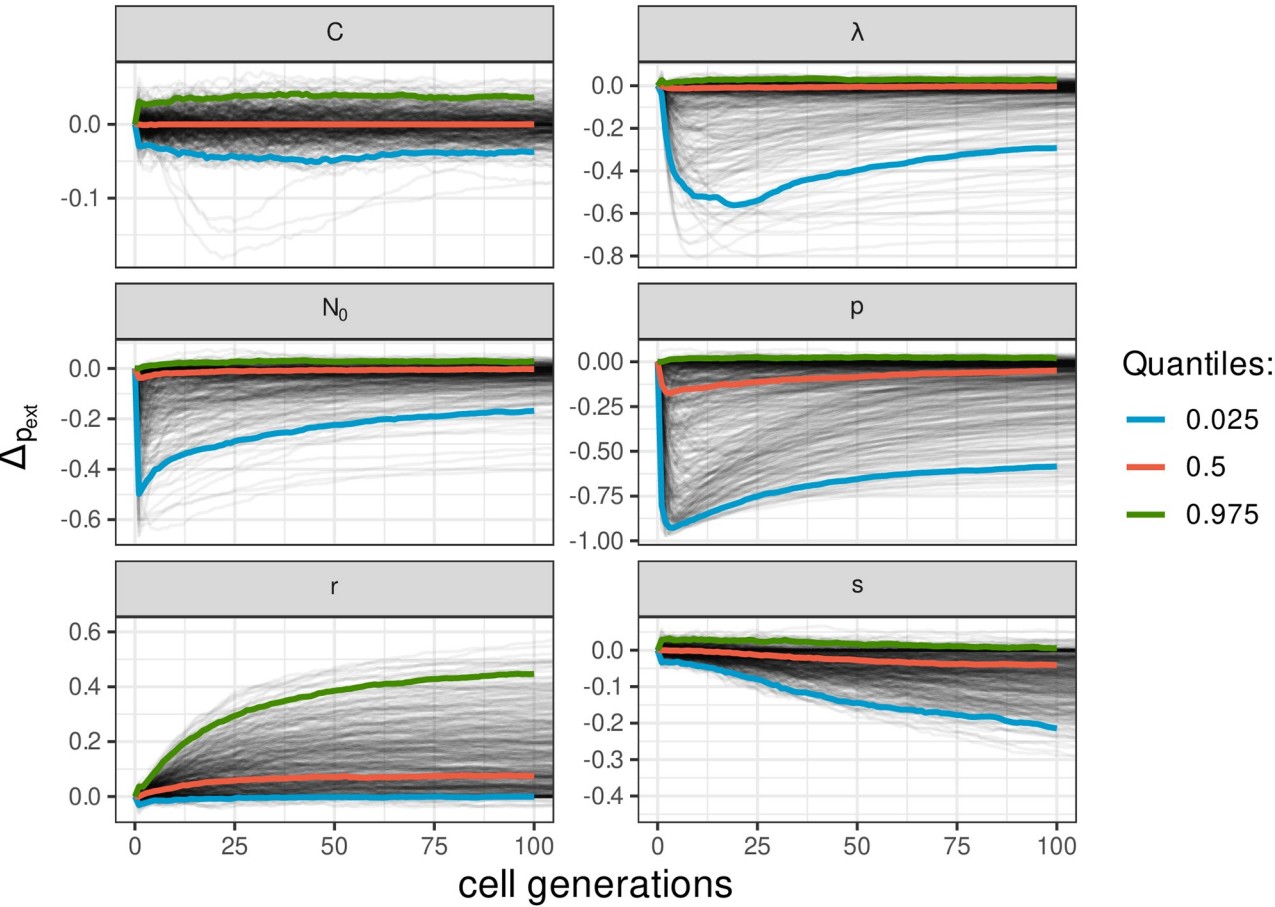

**Fig 1. Difference in the cumulative probability of extinction for two sets of parameters that only differ by one value.** On each panel, the title indicates which parameter varies between the two sets. In each of this panel, we subtract the probability of extinction for the lowest of the two varying parameters to the probability of extinction of the highest of the two. We find consistent results as theory prediction: the probability of extinction decreases with $\lambda$, $p$, $s$, $N_0$ and increase with $r$. The viral capacity in cell ($C$), has little to no effect on the probability of infection. Line colors show the median (in red) and the first and third quartiles (in blue and green).

As explained in the Methods, the numerical simulations slightly deviate from the analytical framework because the number of episomes per cell is assumed to be limited to $C$. However, we expect most of these results to hold in the general framework because we focus on a maximum time scale of 100 cell generations. We observe more deviations on that time interval when $C$ is small and when the intracellular regime is slightly supercritical (further details can be found in S1 Text and S5 Fig).

### Inter-cellular stochasticity

When omitting intracellular stochasticity, theory allows us to predict that the branching process will go extinct with probability $p_{\text{ext}} = \min(r/s, 1)$, which we confirm with numerical simulations, (S4 Fig). Besides, we also find consistent results with theory as the estimated $p_{\text{ext}}$ decreases (resp. increases) with $s$ (resp. $r$), in Fig 1.

As intracellular stochastic processes only increase the risk for the infection to go extinct, we can use as the lower bound for the cumulative probability of extinction of our general process the cumulative probability of extinction for the sole inter-cellular process, the proof is detailed in S1 Text. Generally speaking, no matter how rapidly the virus replicates at the intracellular

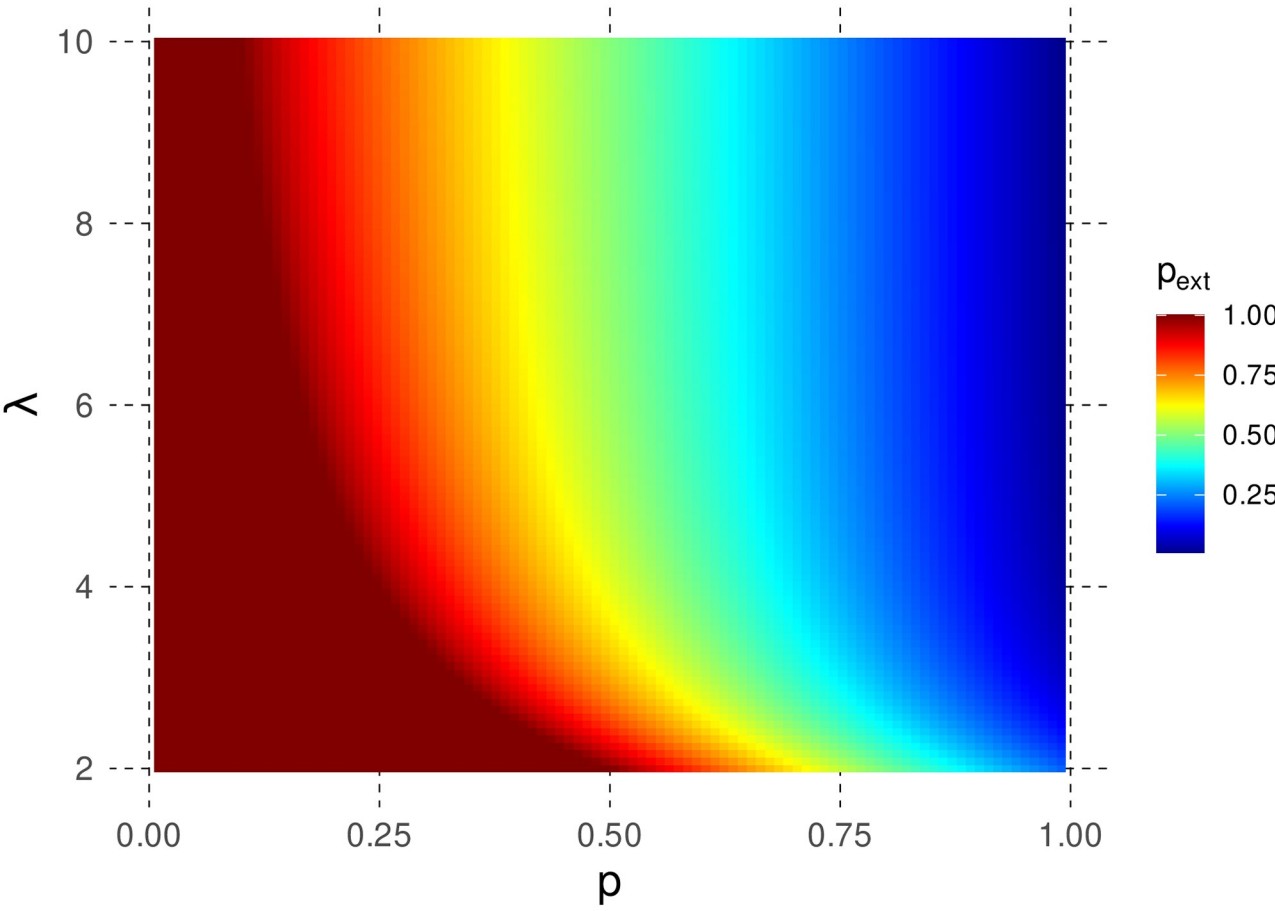

**Fig 2. Effect of episome replication ($\lambda$) and asymmetry in episome partitioning ($p$) on the cumulative probability of extinction ($p_{\text{ext}}$).** We here assume a Poisson scenario. $p_{\text{ext}}$ decreases with both $\lambda$ and $p$, and reaches a threshold $p_{\text{ext}} = 1 - p$ for a given $p$ when $\lambda$ increases. This is consistent with the fact that if $\lambda$ is sufficiently high, the main source of extinction is the first cell division of a stem cell containing 1 episome.

level, its spread is bounded by the cellular dynamics of the host tissue. Therefore, we have

$$\mathbb{P}[X_\infty = 0 | \Gamma_0 = 1] \geqslant \min\left(\frac{r}{s}, 1\right) \tag{5}$$

Globally, these results suggest that symmetric divisions can favor the persistence of the virus depending on the local context of the tissue. In particular, when $s > r$, e.g. when the epithelium switches from a maintenance behavior to a repair behavior [13, 14], symmetric divisions appear to favor the spread of the virus, i.e. its persistence in the tissue. Additionally we can notice in Eq 2 that if the inter-cellular process is critical (i.e. $r = s$), then the general process is subcritical. Indeed, under such circumstances we have

$$m := \mathbb{E}[\phi] = 1 - \psi \leqslant 1 \Rightarrow p_{\text{ext}} = 1.$$

In Fig 3, the proportion of the variance explained by symmetric divisions increases over time. Indeed, on average the first symmetric division occurs after $(1/(r + s))$ cell generations. Hence, the early phase of the infection is mostly governed by intracellular aspect. But later in the infection, if the time between two symmetric events is sufficiently large, the intracellular dynamic has almost surely converged to one of the two states $\{0, C\}$. Under such circumstances, the inter-cellular aspect is the only one acting on the dynamic of the infection. We

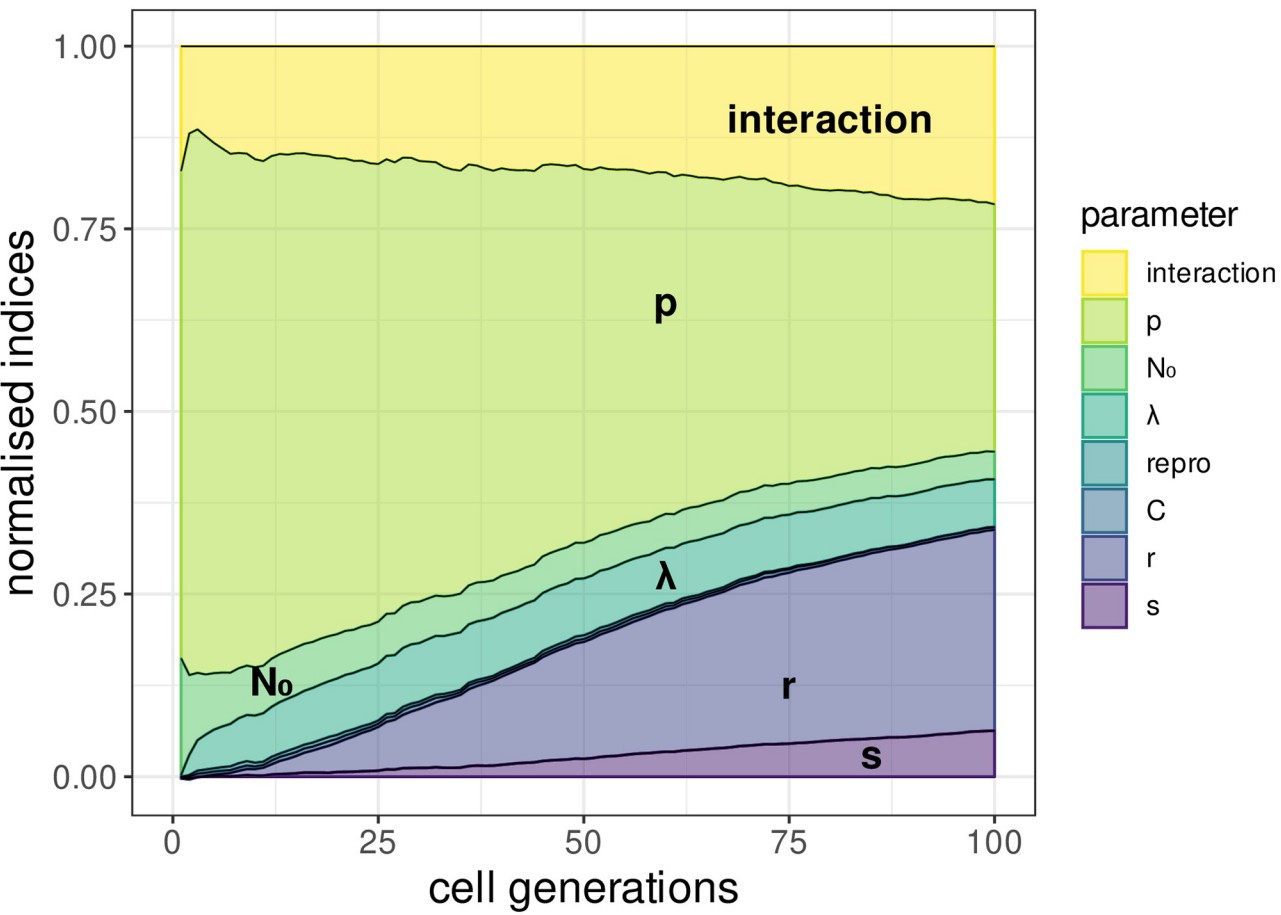

**Fig 3. Variance-based sensitivity analysis for the cumulative probability of extinction ($p_{ext}$).** The normalized stacked first-order sensitivity indices ($\text{Var}_i/\text{Var}_{p_{ext}}$, with $\text{Var}_{p_{ext}}$ the total variance of $p_{ext}$) are displayed for all time steps and all parameters. The intracellular aspect dominates early in the infection. The importance of inter-cellular aspect increases over time, mostly through the probability of symmetric divisions in two differentiated cells. Intracellular viral capacity $C$ and type of reproduction (e.g. Dirac or Poisson distributed) are barely visible due to their small effect on the variability of the outputs.

observe a significant difference in the proportion of variance explained by the parameter $r$ compared with $s$, the latter having a smaller slope. This difference is directly induced by the intracellular stochasticity that can terminate infections prematurely.

## Initial conditions

We then assess the contribution of the initial number of episomes and their distribution in one or more stem cells on the probability of extinction. Let us assume the branching process to start with a total of $X_0 = v \times \omega$ episomes, $(v, \omega) \in \mathbb{N}^2, v > \omega$. We then look for the optimal distribution of these episomes from the virus' perspective. For this, we evaluate the probability of extinction under two extreme scenarios: i) $v$ cells each containing $\omega$ episomes, or ii) $\omega$ cells each containing $v$ viral copies. From branching theory, we know that for the inter-cellular level, the probability of extinction when starting with $\alpha \in \mathbb{N}$ identical and independent infected cells is equal to the probability of extinction of one cell raised to the power $\alpha$.

Therefore, we only need to compute the probabilities of extinction for two sets of parameters that are identical, except for the number of episomes and number of infected stem cells at time $t = 0$, that are switched. We use numerical simulations to evaluate those probabilities,

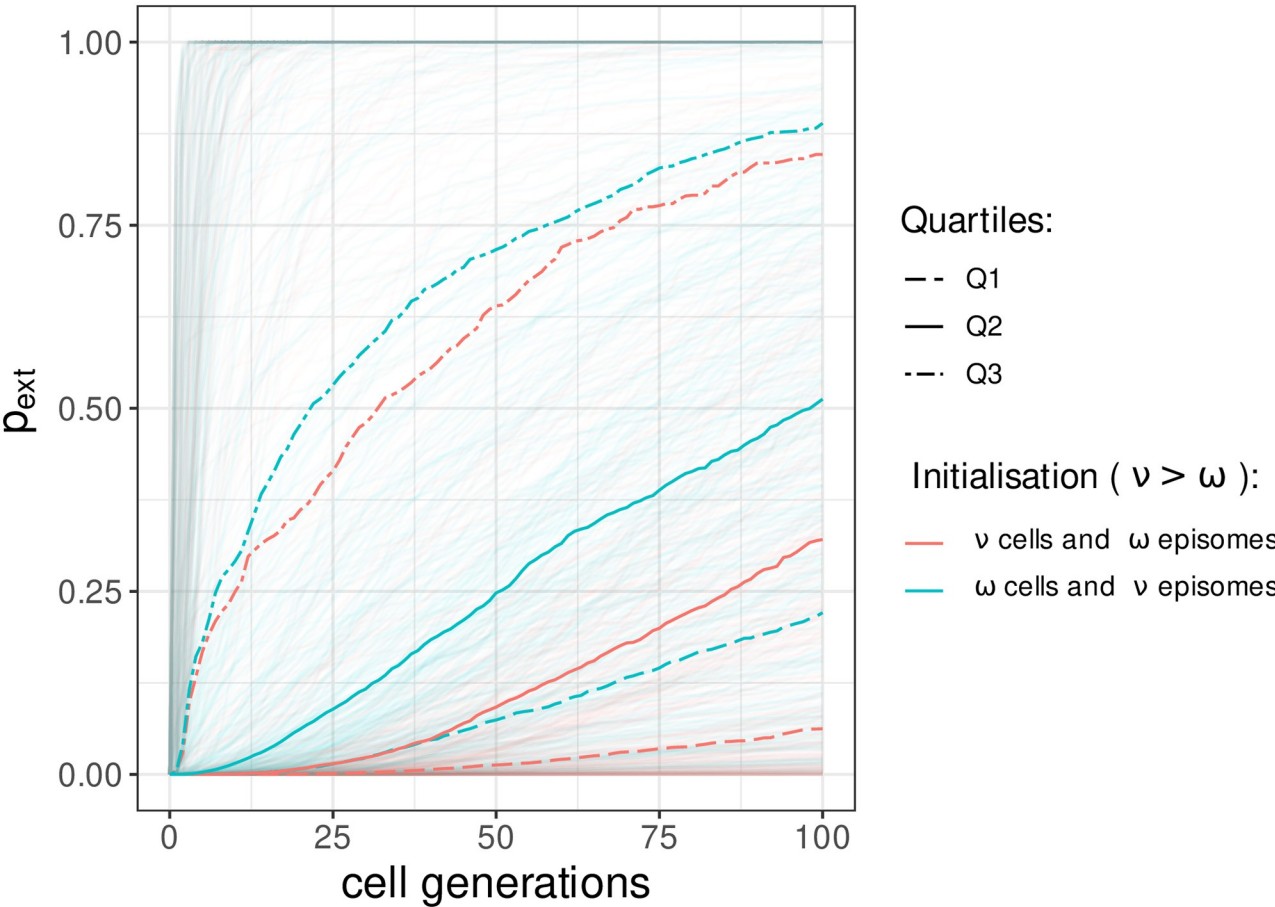

**Fig 4. Estimation of the cumulative probabilities of extinction and their quartiles for the two extreme scenarios.** In red are displayed the estimations when the $N_0 \times \Gamma_0$ initial viral copies are allocated in $\nu$ cells, each containing $\omega$ episomes whereas in blue, the episomes are distributed in $\omega$ cells, each containing $\nu$ copies ($\nu > \omega$). The three quartiles are lower for the first scenario, thus indicating a lower risk of extinction when the copies are split in the maximum number of cells possible.

S6 Fig shows the estimations of $p_{ext}$ for six nearly-identical sets of parameters that only vary with respect to $N_0$ and $\Gamma_0$.

To determine which scenario favors virus persistence we compare the cumulative probabilities of extinction for each of pairs of parameters that only differ by the initial arrangement of the viral copies. Fig 4 shows the quartiles for the two extreme scenarios. This indicates a benefit for the virus to spread its copies to the maximum number of stem cells at the start of the infection, the bigger the difference between $\nu$ and $\omega$ the more significant this benefit is.

## Discussion

As immune mechanisms fail to completely explain the problem of spontaneous clearance in natural infections, Ryser [17] has suggested random events in the type of cell divisions could induce extinction and thus complement explanations based on immunity. Following this approach, we refine the model initially exposed by Ryser [17] and developed a model that also accounts for the risk of stochastic extinction due to virus partitioning. We hypothesized that an HPV infection could face extinction due to the sole effects of random processes occurring both at the intra- and inter-cellular level. Using a stochastic model comprised of two nested

branching processes and numerical simulations, we find that random allocation of episomes during cell divisions drastically limit the persistence in the early phase of the epidemic. Later, if episomes persisted in some infected cells, randomness in the type of cell divisions can also lead to extinction, even though depending on the local context, symmetric divisions can also favor the spread of the virus in the tissue (when $s > r$). Notably, it was shown that the same cells can exhibit different proliferative patterns, switching from a balanced mode to an expanding mode if the tissue is wounded, and back to balanced mode once confluence is reached back [13, 14]. While the precise molecular mechanisms underlying those shifts are still unknown, studies suggest that actin signaling could play a key role in the fate of the keratinocytes [14]. The importance of each aspect varies throughout the infection, and more generally as HPV viral copies increase in number, the role of stochasticity in the explanation of extinction fades. Our results show the first stages of the infection matter a lot for the persistence of the virus, which is particularly important for viruses like HPV that have a low number of copies initially [18].

Another outcome of our model is that many infections by HPV might clear naturally in a matter of weeks/months before even being noticed by the host immune system since episomes remain in small number in the basal layer. Adding to the pioneering work of Ryser [17] to highlight the role of stochastic processes in the clearance of HPV infections, we show that intracellular stochasticity greatly impacts extinction probability, especially in the early stages of the infection. Combining the two processes enables us to reach substantial clearance rates without involving immune response.

These results could explain why the antibody production against the virus is low after natural infections [10, 18], as the infection might fade randomly within the first divisions of the initially infected cells. HPVs might have evolve to limit pressures exerted by randomness during the first cell divisions by amplifying the viral copies in the early phase of the epidemic. Notably in the case of HPV16, the E1 protein seems to play a decisive role in the establishment of a persistent infection until the number of episomes per cell reaches host capacity but the same protein was dispensable for the maintenance of the infection. Our model corroborates these observations: we show that the early phase of the infection can be detrimental to the infection, E1 protein could favor the maintenance of the infection either by acting on the probability to be distributed to the daughter stem cell ($p$ parameter) or promote episome amplification ($\lambda$ parameter). Upon reaching host cell capacity, even if the intracellular regime is critical, it is very unlikely for the virus to go extinct, hence E1 protein is dispensable during maintenance phase of the infection [38].

Papillomaviruses have evolved over million years infecting various animals' epithelium [39]. From this long coevolution between the virus and its host, a large diversity of evolutionary strategies have emerged. For instance, HPV genotypes have developed specific host- and tissue-tropism that manifest through distinct pathologies [40–42]. HPV research mostly focuses on the Alpha genus since it contains all high-risk HPVs (HR-HPVs) that are susceptible to induce various forms of cancer in the genital areas and upper aerodigestive area [2, 43–45]. The Alpha genus also contains genotypes that cause benign infections and are classified as low-risk HPVs (LR-HPVs). Although benign, LR-HPV infections can cause hyperproliferative lesions such as genital warts. These HPVs have been less studied, compared to HR-HPV, despite accounting for a bigger number of types [46]. The distinct evolution of LR and HR-HPVs, which illustrates the persistence—virion production trade-off, has been investigated in the literature [47, 48]. The role of the oncogenic proteins E6 and E7 have been put forward to explain the capacity of the HR-HPV to induce cellular immortalization and cellular progression towards cancer development [49, 50]. However, it is unclear how these differences could impact the early phases of the infection and interfere with the stochastic pressures acting on the virus.

Globally our results suggest that remaining with a low copy number is a way to evade immunity [51], but such a strategy is risky for the virus and is subject to strong stochastic forces that limit the persistence of the infection. Viruses committed to such strategy might alleviate these pressures exerted by randomness using proteins in the very early phase of the infection, as the E1 protein for HPV16 [38]. Unfortunately, these proteins might not be the most adequate targets for antiviral drugs as studies showed that the role of these are crucial in the very early steps of the infection but lose importance once cell carrying capacity is reached [38]. Therefore, we might not be able to detect HPV sufficiently early to inhibit E1 protein to prevent infection persistence.

Ryser [17] assumed a balance between the two types of symmetric divisions, thus restraining the inter-cellular dynamic to the critical regime. We allow the possibility for the tissue to deviate from that assumptions. Notably, we allow the probability of symmetric divisions in two stem cells to exceed the probability of symmetric divisions in two differentiated cells as keratinocytes can alternate between two modes of proliferation depending on local context. If local confluence is removed by scratch injury, cells switch to the expanding mode, characterized by an excess of cycling cells production, until confluence is reached again, after which they switch back to a balanced mode where similar proportions of proliferating and differentiating cells are produced, thus generating population asymmetry maintaining epidermal homeostasis [14]. The two modes of proliferation could differ by the rates of cell divisions or the probability of division in two stem cells (parameter $s$), or both. Our model predicts that HPVs greatly benefit from the expanding mode, which echoes previous empirical work on the link between HPV persistence and wound healing responses [16, 52–54].

To further calibrate the models and delineate the true role of stochasticity, a comparison with longitudinal follow-ups of women recently infected by any type of HPV or presenting chronic latent infections would be usefull. Ideal time windows between two measurements should not exceed 1 month. Besides, we should perhaps reconsider how we label infections that cleared in less than 1 month. Many follow-ups treat them as "carriage", i.e. HPV from the partner and not a "true" infection but perhaps this underestimates the number of non-persistent infections. Overall to decipher this issue, we need accurate infection duration distributions over short time scales, ideally with virus loads data.

Comparing longitudinal follow-ups data and numerical simulations would require clarifying the estimation of the time between two cell divisions. We rely on old measures [25] to set the upper bound of the number of cell generations to compute in our three years simulations. This work estimates the rate of divisions around [0.03, 0.07] day$^{-1}$. These results are consistent with more recent work on the kinetics of epidermal maintenance stating an overall division rate of 1.1 per week [55]. To fit his model to follow-ups data, Ryser [17] estimated the rate of division using proxies (height of the stratified squamous epithelium expressed in number of cells, renewal time of the tissue, fraction of proliferative cells in the basal layer) and found a much higher division rate [0.29, 1] day$^{-1}$. Improving our knowledge of within-host processes would be of primary interest to compare theoretical and experimental approaches.

## Supporting information

**S1 Fig. Diagram of the stochastic model of HPV-infected cells division.** Stem cells in the basal layer divide mostly asymmetrically. Rare symmetric divisions occur that either yield two stem cells (with probability $r$) or two differentiated cells (with probability $s$). These can drastically impact the dynamic of the infection. Upon asymmetric division, viral copies are allocated randomly in the two daughter cells. Each episome is allocated independently to the daughter

stem cell with probability $p$ or else goes into the differentiated cell that migrates towards the epithelium surface. These cells do not participate in the persistence of the infection. Copies allocated to the stem cell are then amplified in $\lambda$ copies on average. This random process is repeated every asymmetric division. The inter-cellular diagram is largely inspired from Fig 1 in reference [17].
(TIF)

**S2 Fig. Stochastic trajectory of the dynamic of the number of episomes over time.** On the left panel (A) we displayed the dynamics inside each cell lineage over time. On the right panel (B) we plot the variation in the total number of episomes. We fixed the value of the parameters as follows: $p = 0.5$, $\lambda = 2$, $s = r = 0.02$, $N_0 = 10$ and $C = 200$. The episomes amplification is fixed and happens after cell divisions. On panel (A), each color represents a cell lineage (more information on cell lineage labeling can be found in S1 Text).
(TIF)

**S3 Fig. Estimated cumulative probability of extinction ($p_{ext}$ with finite intra-host capacity (plain dots, ribbons correspond to 95% confidence intervals) and according to theory (dashed lines).** In most cases, the estimations of $p_{ext}$ under constrained host capacity do not diverge from theoretical predictions on the time window considered. Deviations emerge when the intra-host regime is slightly supercritical and/or the intra-host capacity is limited ($<50$). The colors indicate the row of the parameter matrix LHS_intra (see S1 Text) used to obtain the cumulative probability of extinction.
(TIF)

**S4 Fig. Estimated cumulative probability of extinction ($p_{ext}$ with finite intrahost capacity (plain dots, ribbons correspond to 95% confidence intervals) and according to theory (dashed lines).** We observe no significant differences between estimations and theoretical predictions. The colors indicate the row of the parameter matrix LHS_inter (see S1 Text) used to obtain the cumulative probability of extinction.
(TIF)

**S5 Fig. Effect of episome replication ($\lambda$) and asymmetry in episome partitioning ($p$) on the cumulative probability of extinction ($p_{ext}$).** We here assume a Dirac scenario. $p_{ext}$ decreases with both $\lambda$ and $p$, and reaches a threshold $p_{ext} = 1 - p$ for a given $p$ when $\lambda$ increases. This is consistent with the fact that if $\lambda$ is sufficiently high, the main source of extinction is the first cell division of a stem cell containing 1 episome.
(TIF)

**S6 Fig. Estimations of the cumulative probability of extinction ($p_{ext}$) for two sets of parameters that only differ by the allocation of the viral copies at the start of infection.** We display in red the estimations of $p_{ext}$ in the scenario where the initial few viral copies are spread in more cells and in blue the scenario where more viral copies are spread in less cells. The solid lines indicates the estimations of $p_{ext}$ and the ribbon the 95% confidence intervals. The cumulative probability of extinction is generally lower in the first scenario compared to the latter. Thus it is more beneficial for the virus to spread its copies in the maximum number of cells at the the start of the infection. On each facet, the title indicates the row of the parameter matrix LHS_A (see S1 Text).
(TIF)

**S1 Text. Mathematical details and extra information on the simulation procedure.**
(ZIP)

 

## Acknowledgments

The authors acknowledge the IRD itrop HPC (South Green Platform) at IRD Montpellier for providing HPC resources that have contributed to the research results reported within this paper.

## Author Contributions

**Conceptualization:** Thomas Beneteau, Christian Selinger, Mircea T. Sofonea, Samuel Alizon.

**Formal analysis:** Thomas Beneteau, Christian Selinger, Mircea T. Sofonea, Samuel Alizon.

**Methodology:** Thomas Beneteau, Christian Selinger, Mircea T. Sofonea, Samuel Alizon.

**Supervision:** Christian Selinger, Mircea T. Sofonea, Samuel Alizon.

**Visualization:** Thomas Beneteau.

**Writing – original draft:** Thomas Beneteau.

**Writing – review & editing:** Thomas Beneteau, Christian Selinger, Mircea T. Sofonea, Samuel Alizon.

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
