## [Decision Letter · Decision Letter 0]

7 Jul 2021

Dear Mr. BENETEAU,

Thank you very much for submitting your manuscript "Episome partitioning and symmetric cell divisions: quantifying the role of random events in the persistence of HPV infections" for consideration at PLOS Computational Biology. As with all papers reviewed by the journal, your manuscript was reviewed by members of the editorial board and by several independent reviewers. The reviewers appreciated the attention to an important topic. Based on the reviews, we are likely to accept this manuscript for publication, providing that you modify the manuscript according to the review recommendations.

Sincerely,

Roger Dimitri Kouyos

Associate Editor

PLOS Computational Biology

Rob De Boer

Deputy Editor

PLOS Computational Biology

[LINK]

Reviewer's Responses to Questions

**Comments to the Authors:**

Reviewer #1: I carefully read the manuscript “Episome partitioning and symmetric cell divisions: quantifying the role of random events in the persistence of HPV infectionsn”, which develop a stochastic mathematical model of HPV dynamics with an explicit description of the intracellular level. This is a well organized and clearly written paper containing some valuable results.I have checked the code that underpins the findings, which is well commented and explained. In general, I suggest accepting the manuscript with minor revision.

However，There are some shortcomings that need to be addressed：

1.In the Author summary and the Introduction section, the logic of some sentences is not easy to understand, for example, sentences 4-5 in the author's summary: “Hence for a given HPV infection, the risk of progression towards cancerous status is low. Unfortunately, the prevalence of HPVs is very high, which makes it a major public health issue.”

2.The figures in the manuscript are a little oversized, but not of high quality, so they are not particularly clear.

3. Why should the Acknowledgments be added between S6 and S7 instead of after S7?

Reviewer #2: Reviewer’s comments on the paper entitled “Episome partitioning and symmetric cell divisions: quantifying the role of random events in the persistence of HPV infections” by Thomas et al.

This paper deals with the mathematical modelling of the Human Papillomaviruses (HPV), one of the most prevalent sexually transmitted infections (STI). The author developed a stochastic mathematical model of HPV dynamics with an explicit description of the intracellular level. They showed that the random partitioning of virus episomes upon stem cell division and the occurrence of symmetric divisions greatly affect viral persistence. Their findings gives a better understand the relative importance of stochasticity and immunity in HPV infection clearance.

Numerical simulation is also performed to verify the analytical findings. This manuscript is written well and seems complete in every aspect. However, the reviewer feels that the authors should discussed some recent studies

1. Chatterjee, A. N, Ahmad, B., “A fractional-order differential equation model of COVID-19 infection of epithelial cells”, Chaos, Solitons & Fractals , V 147 (2021), 110952.

2. Mondal, Jayanta, Piu Samui, and Amar Nath Chatterjee. "Optimal control strategies of non-pharmaceutical and pharmaceutical interventions for COVID-19 control." Journal of Interdisciplinary Mathematics (2020): 1-29.

3. Chatterjee, Amar Nath, Fahad Al Basir, and Yasuhiro Takeuchi. "Effect of DAA therapy in hepatitis C treatment—an impulsive control approach." Mathematical Biosciences and Engineering 18, no. 2 (2021): 1450-1464.

4. Chatterjee, A.N. and Al Basir, F., 2020. A model for sars-cov-2 infection with treatment. Computational and mathematical methods in medicine, 2020.

5. Chakraborty, Sudip, and Priti Kumar Roy. "Therapeutic control of HPV associated cervical cancerous cell and its possible extinction." Nonlinear Studies 26, no. 2 (2019).

6. Chakraborty, S., Xianbing Cao, S. Bhattyacharya, and P. K. Roy. "The Role of HPV on cervical cancer with several functional response: a control based comparative study." Computational Mathematics and Modeling 30, no. 4 (2019): 439-453.

7. Chakraborty, Sudip, Xue-Zhi Li, and Priti Kumar Roy. "How can HPV-induced cervical cancer be controlled by a combination of drug therapy? A mathematical study." International Journal of Biomathematics 12, no. 06 (2019): 1950070.

The overall manuscript seems fine and it may be accepted if authors incorporate the above points/query in the revised manuscript.

Reviewer #3: The review is uploaded as an attachment

**Have the authors made all data and (if applicable) computational code underlying the findings in their manuscript fully available?**

Reviewer #1: Yes

Reviewer #2: None

Reviewer #3: Yes

PLOS authors have the option to publish the peer review history of their article (what does this mean?). If published, this will include your full peer review and any attached files.

Reviewer #1: No

Reviewer #2: No

Reviewer #3: No

Figure Files:

Data Requirements:

Reproducibility:

References:

---

## [Editor Report · Decision Letter 1]

16 Aug 2021

Dear Mr. BENETEAU,

We are pleased to inform you that your manuscript 'Episome partitioning and symmetric cell divisions: quantifying the role of random events in the persistence of HPV infections' has been provisionally accepted for publication in PLOS Computational Biology.

Best regards,

Roger Dimitri Kouyos

Associate Editor

PLOS Computational Biology

Rob De Boer

Deputy Editor

PLOS Computational Biology

---

## [Editor Report · Acceptance letter]

31 Aug 2021

PCOMPBIOL-D-21-00650R1 

Episome partitioning and symmetric cell divisions: quantifying the role of random events in the persistence of HPV infections

Dear Dr BENETEAU,

I am pleased to inform you that your manuscript has been formally accepted for publication in PLOS Computational Biology. Your manuscript is now with our production department and you will be notified of the publication date in due course.

With kind regards,

Andrea Szabo
